# Establishment and Characterization of *hTERT* Immortalized Hutchinson–Gilford Progeria Fibroblast Cell Lines

**DOI:** 10.3390/cells11182784

**Published:** 2022-09-06

**Authors:** Haihuan Lin, Juliane Mensch, Maria Haschke, Kathrin Jäger, Brigitte Köttgen, Jens Dernedde, Evelyn Orsó, Michael Walter

**Affiliations:** 1Charité—Universitätsmedizin Berlin, Corporate Member of Freie Universität Berlin, Humboldt-Universität zu Berlin, and Berlin Institute of Health, Institute of Laboratory Medicine, Clinical Chemistry and Pathobiochemistry, 13353 Berlin, Germany; 2Institute of Clinical Chemistry and Laboratory Medicine, Rostock University Medical Center, 18057 Rostock, Germany; 3Institute of Clinical Chemistry and Laboratory Medicine, University Hospital Regensburg, 93053 Regensburg, Germany

**Keywords:** Hutchinson–Gilford progeria syndrome (HGPS), telomerase, telomere length, cell senescence, β-galactosidase

## Abstract

Hutchinson–Gilford progeria syndrome (HGPS) is a rare premature aging syndrome caused by a dominant mutation in the *LMNA* gene. Previous research has shown that the ectopic expression of the catalytic subunit of telomerase (*hTERT*) can elongate the telomeres of the patients’ fibroblasts. Here, we established five immortalized HGP fibroblast cell lines using retroviral infection with the catalytic subunit of *hTERT*. Immortalization enhanced the proliferative life span by at least 50 population doublings (PDs). The number of cells with typical senescence signs was reduced by 63 + 17%. Furthermore, the growth increase and phenotype improvement occurred with a lag phase of 50–100 days and was not dependent on the degree of telomere elongation. The initial telomeric stabilization after *hTERT* infection and relatively low amounts of *hTERT* mRNA were sufficient for the phenotype improvement but the retroviral infection procedure was associated with transient cell stress. Our data have implications for therapeutic strategies in HGP and other premature aging syndromes.

## 1. Introduction

Hutchinson–Gilford progeria syndrome (HGPS) is a genetic disease characterized by premature aging features, affecting one in 4–8 million children [1]. The affected children look normal at birth but begin to slow down growth from one year onwards. Characteristic symptoms appear during the first few years, including growth retardation, thin skin, hair loss, osteoporosis, reduction in subcutaneous fat, and development of coronary heart disease [1]. The average life expectancy of HGPS patients is 14.6 years [2]. Most patients die from stroke or other atherothrombotic complications due to extensive and abnormal atherosclerosis [3,4].

Circa 90% of HGPS patients are characterized by a *de novo* 1824 C to T mutation within exon 11 of *LMNA*, which activates a cryptic splice site and results in the deletion of 50 amino acids near the C-terminus of the *LMNA* product prelamin A [5,6]. The originating abnormal protein is called “progerin”, which retains the -CAAX motif and is farnesylated and carboxymethylated. The function of the -CAAX motif is to provide signals for the addition of farnesyl groups to prelamin A [7]. The farnesylated prelamin A is located in the nuclear membrane, where it is cleaved by the zinc metalloproteinase ZMPSTE24 to cut off the final 15 amino acids at the C-terminus, including the farnesyl group. Finally, the mature lamin A inserts into the nuclear layer [8,9]. However, progerin lacks the second site for endoproteolytic cleavage. The permanent farnesylation and carboxymethylation cause progerin to remain anchored in the nuclear membrane. This accumulation of progerin destroys the underlying lamina structure, leads to errors in the nuclear envelope, protein localization, and multiple changes in chromatin structure and gene transcription [6,10,11], whereby the severity of HGPS depends on the amount of progerin relative to prelamin A [12,13].

HGPS fibroblasts are characterized by accelerated telomere attrition, which was proposed to play a key role in HGPS pathology [14,15,16]. Human telomeres form a protective structure at the chromosome ends, composed of GT-rich tandem repeats and associated with various telomere binding proteins [17]. This telomere structure comprises a cap and helps to maintain the integrity of chromosomes and prevents the recognition of chromosomes as broken or damaged DNA. Due to the end replication problem (DNA polymerase fails to fully replicate the DNA lagging strand), the telomeres are progressively shortened at every cell division cycle. Telomere erosion will eventually trigger a DNA damage response, leading to irreversible growth arrest and senescence [18,19,20]. Telomerase, a reverse transcriptase, can potentially circumvent telomere shortening by adding telomeric repeats at the 3’ ends of chromosomes [21]. The two critical components of telomerase are human telomerase reverse transcriptase (*hTERT*) and telomerase RNA (TR). TR is available in excess, but *hTERT* has low abundance in most tissues. Most human tissues except stem cells do not express sufficient amounts of telomerase to compensate for natural telomere loss. Therefore, telomeres gradually shorten as cells divide, leading to replicative senescence.

Previous studies have shown that ectopic expression of *hTERT* may improve the HGPS phenotype in cultivated fibroblasts [22,23,24,25]. However, the interrelationships among the *hTERT* mRNA induction, the degree of telomere elongation and the cellular phenotype improvement are less clear. In this study, we compared the amount of *hTERT* mRNA, the degree of telomere elongation and the reduction in cellular stress levels before and after immortalization with *hTERT* control cells and HGPS fibroblasts.

## 2. Materials and Methods

### 2.1. Cell Culture

Patient-derived HGP fibroblasts (HGADFN 003, HGADFN 127, HGADFN 164, HGADFN 178 and HGADFN 188, shortened HGP 003, HGP 127, HGP 164, HGP 178, and HGP 188) with a single-point mutation (C->T) in exon 11 of *LMNA* were provided by the Progeria Research Foundation Cell and Tissue Bank. Professor Evelyn Orsó provided healthy control fibroblast cells (N14, N16, N17, N18, and N20) from the University Hospital Regensburg (Table 1). Cells were cultured in Dulbecco’s modified Eagle medium (DMEM) High Glucose (4.5 g/L) with L-Glutamine, without Pyruvate (GIBCO, Thermo Fisher Scientific, Waltham, MA, USA, Cat. 41965) supplemented with 10% fetal bovine serum (FBS) (GIBCO, Thermo Fisher Scientific, Waltham, MA, USA, Cat. 10270), 100 U/mL penicillin and 100 µg/mL streptomycin (Biochrom, Berlin, Germany, Cat. A2213). All cells were grown at 37 °C in 5% CO_2_. Cellular growth rates were graphed as growth curves at 100% confluence by the changes in passage doublings (PDs) related to the days in culture.

### 2.2. Immortalization of Fibroblasts Using the Packaging Cell Line PA 317-hTERT

All fibroblasts underwent infection procedures with a virus supernatant containing *hTERT*-pBabe Puro vector from the packaging cell line PA317 in the presence of 4 μg/mL polybrene (Santa Cruz Biotechnology, Dallas, TX, USA, Cat. SC-134220) for two days. The vectors and packaging cell line were a gift from Dr. Woodring Wright (UTSW Medical School, Dallas, TX, USA). Following the virus infection, 1 μg/mL puromycin (GIBCO, Thermo Fisher Scientific, Waltham, MA, USA, Cat. A11138) was used for four days to remove uninfected cells and acquire virus-infected cells. After the selection, cells were maintained in serial subcultures for additional 50 PDs and harvested per each passage for further study. We calculated the growth curves as described above.

### 2.3. RNA Extraction and RT-PCR

Total cellular RNA was extracted from cells by a standard extraction kit (Macherey-Nagel NucleoSpin RNA, Düren, Germany, Cat. 740955). The obtained RNA was quantified using a NanoDrop (ND 1000) spectrophotometer. A total of 1 μg RNA from each sample was transcribed into cDNA using random primers (Promega, Madison, WI, USA, Cat. C1181) and reverse transcriptase (M-MLV Reverse Transcriptase, RNase (H-), Point Mutant, Promega, Madison, WI, USA, Cat. M3682). Then, cDNA samples (5 ng/µL) were analyzed by real-time quantitative PCR on a BioRad CFX384 real-time C1000 PCR Cycler in triplicates.

*hTERT* mRNA expression in control and Progeria cell lines was measured by real-time quantitative PCR on a BioRad CFX384 real-time C1000 PCR Cycler. For each PCR run, a master mix was prepared on ice with one unit of TaqMan buffer, 0.2 mM each dNTP, 150 nM each primer, 0.75% Sybr Green (Roche, Basel, Switzerland, Cat. 04673514001), and one unit of Titanium Taq DNA Polymerase (Takara Bio Europe Saint-Germain-en-Laye, France, Cat. 639209). For the standard curve, 2.5 µL of cDNA (started from 100 ng, and diluted additional six points according to the ratio of 1:3) from one successfully immortalized control cell line (731, male, five months, no molecular defect) were added to 7.5 µL of the PCR master mix with a dilution coefficient of 3. For the unknown samples, 2 µL of each cDNA sample (5 ng/µL) were added to 8 µL of the PCR master mix. All samples (unknown or standard) were run in triplicates. We analyzed the none template control (water) in the same way. Thermal cycling conditions included an initial denaturation step at 95 °C for 10 min, followed by 50 cycles at 95 °C for 15 s and 1 min at 60 °C with the signal acquisition. The sequences for the *hTERT* primer were: forward 5’-CCG ATT GTG AAC ATG GAC TAC GT-3’, reverse: 5’-CGT AGT TGA GCA CGC TGA ACA G-3′. The sequences for the GAPDH primer were: forward 5’- CTC TGC TCC TCC TGT TCG AC -3’, reverse: 5’- GCG CCC AAT ACG ACC AAA TC-3′.

The *hTERT* target nucleic acid is quantified by measuring Ct (threshold cycle) and using a standard curve to determine the starting target nucleic acid quantity. Glyceraldehyde 3-phosphate dehydrogenase (GAPDH, endogenous control) was analyzed as the endogenous mRNA control, and each sample was normalized based on its GAPDH content. mRNA expression was described as a fold-change in the reference cell’s basal value in two ways. One is “absolute” mRNA expression, which was calculated as a surrogate marker by comparison to the basal value of one “standard” primary control cell (N17 at low PD). Another is relative mRNA expression measured by comparison to the basal value of its respective primary cell at low PD.

### 2.4. DNA Extraction and Telomere Length Measurement

According to the manufacturer’s protocol, the genetic DNA of fibroblasts was extracted using a standard extraction kit (DNeasy Blood & Tissue Kit, Qiagen, Hilden, Germany, Cat. 69504). The concentration of DNA samples was analyzed using a NanoDrop (ND 1000) spectrophotometer. The telomere length (TL) was measured using a monochrome multiplex quantitative PCR (MMQPCR) method described by Cawthon [26]. This method can amplify telomeres and single-copy genes (reference) in a single reaction well at different temperatures. The ratio of telomere (T) to single-copy gene (S) content (T/S ratio) was used as a relative measure of TL and expressed in arbitrary units. The sample DNA (20 ng/µL) and standard DNA (diluted 1:3 seven times starting from 164.29 ng/µL) were measured in technical triplicates, each on a separate plate. The no-template control (water) and the positive control (human leukemia cell line 1301) were prepared in duplicate and run on each plate. We run the assay on a BioRad CFX384 real-time C1000 thermal cycler with the following thermal cycling conditions: 1 cycle 15 min at 95 °C; 2 cycles of 15 s at 94 °C, one cycle 15 s at 49 °C; 40 cycles of 15 s at 94 °C, one cycle 10 s at 62 °C, one cycle 15 s at 72 °C with T signal acquisition, 10 s at 85 °C, and 15 s at 89 °C with the signal acquisition. The primers of the telomere (T) used were: telg, 5′-ACA CTA AGG TTT GGG TTT GGG TTT GGG TTT GGG TTA GTG T–3’ and telc, 5′-TGT TAG GTA TCC CTA TCC CTA TCC CTA TCC CTA TCC CTA ACA–3’. The primer sequences of the single-copy gene (S) used were albu: 5’-CGG CGG CGG GCG GCG CGG GCT GGG CGG AAA TGC TGC ACA GAA TCC TTG–3’ albd: 5′-GCC CGG CCC GCC GCG CCC GTC CCG CCG GAA AAG CAT GGT CGC CTG TT–3’.

### 2.5. Senescence-Associated β-Galactosidase (SA-β-gal) Quantitative Assay

We used Ronald’s quantitative assay to analyze senescence-associated β-galactosidase (SA-β-gal) activity [27]. In this assay, the SA-beta-gal activity was measured by the conversion rate of 4-methylumbelliferyl-β-D-galactopyranoside (4-MUG) (Sigma-Aldrich, (Sigma-Aldrich, St. Louis, MO, USA, Cat. M1633) to a fluorescent product 4-methylumbelliferone (4-MU) at pH 6.0. We mixed all required reagents and followed the processes described by Ronald [27]. We cultured the cells to 50–70% confluence and obtained the clarified supernatant of the cellular lysate. Then, 100 μL lysis buffer was added to the reaction tube and mixed with 50 μL lysis buffer and 150 μL 2× reaction buffer. A 1.7 mM 4-MUG (Sigma-Aldrich, St. Louis, MO, USA, Cat. M1633; stored at −20 °C) was added immediately before the reaction started. The reaction solution was placed into a 37 °C water bath for three hours. Then, 50 μL of the reaction solution was withdrawn and mixed with 500 μL of stop solution (400 mM sodium carbonate) to stop the enzymatic reaction. Then, 150 μL of the mixed solution (in duplicates) was read using Fluoroscan Ascent FL (Thermo Fisher Scientific, Waltham, MA, USA), at 360 nm excitation and 465 nm emission wavelength. We normalized the SA-β-gal activity by dividing the detected fluorescence by micrograms of total protein. The lysate’s total protein concentration was measured by BCA assay (Thermo Fisher Scientific, Waltham, MA, USA, Cat. 23225) using human albumin as the standard.

### 2.6. Statistical Analysis

Due to a low number of different cell lines, we used nonparametric tests only.

We analyzed the data between the two groups with the Mann–Whitney-U-Test using SPSS statistics 27. Exact p-values are displayed. Bonferroni correction was performed by multiplying p-values with the number of testing (*n* = 3) to reduce type 1 error. A corrected *p*-value of ≤0.016 was considered to indicate a statistically significant difference.

Spearman’s rho correlation coefficient was used to assess the relationship between telomere length and mRNA expression using SPSS statistics 27.

## 3. Results

Previous studies have shown that the ectopic expression of *hTERT* can immortalize human fibroblasts [28,29]. These studies did not report in detail to what extent the telomeres have been elongated, which levels of *hTERT* mRNA were required for telomere elongation, and how many cell cycles were required for immortalization. Therefore, we compared cell growth, total numbers of PDs, *hTERT* mRNA expression levels, and TL kinetics.

### 3.1. Delayed Proliferation Induction after Immortalization Independent of the Amount of Cellular hTERT mRNA

We used retrovirus-mediated transfer to introduce *hTERT* to five HGP fibroblasts (PDs 23–40) and five healthy control fibroblasts (PDs 6–24). Identical cell lines infected with empty backbone vectors were used as negative controls (Table 1).

For comparative analyses, we categorized cells into three groups: cells in a low (L) PD (10–20 PDs in HGP cells, 10–15 PDs in control cells); cells in a medium PD (M) (21–30 PDs in HGP cells, 15–25 PDs in control cells); and cells in a high (H) PD (>30 PDs in HGP cells, >25 PDs in control cells) (Figure 1 and Figure 2). Likewise, we categorized immortalized cells into three groups: cells immediately after immortalization in a low (L) PD between PD 7 and 20; in a medium (M) PD between PD 20 and 30; and in a high (H) PD above PD 47 (Figure 1).

After viral transduction, the proliferation rates displayed a period of stagnation for 50–100 days. Both virus-infected control and virus-infected HGP cells showed increased growth rates following this period of stagnation for at least additional 300 days or 50 PDs without apparent signs of senescence (Figure 1); moreover, a tendency for growth acceleration at higher PDs was found in some cell lines. By contrast, the primary fibroblasts proliferated for a limited number of PDs to a lower maximum PD than the *hTERT* immortalized cells. The control cells with empty vectors grew similar to the primary cells without transduction but senesced earlier in all instances (Figure 1). The HGPS cells with empty vector died a few PDs after infection (Figure 1). Thus, all primary fibroblasts grew to higher PDs than the fibroblasts with empty vectors, suggesting some negative effects of the immortalization procedure *per se*.

High *hTERT* mRNA levels were induced shortly after the infection in both HGP and control cells (Figure 2). The growth rates, however, were not related to the degree of *hTERT* mRNA expression. The *hTERT* mRNA expression of four HGP cell lines (HGP127, HGP164, HGP178, and HGP188) and one control cell line (N18) was reduced at higher PDs, although these cells continued growing at high proliferation rates (Appendix A).

### 3.2. No Positive Correlation between hTERT Expression and TL

Compared to the primary cells at low PD, the mean TLs of *hTERT*-infected cells nearly doubled in control cells. HGP cells also showed markedly increased TLs (Figure 2). The average increase (relative to the medium PD) was 92%, 95%, 135% in control cells, and 45%, 56%, 75% in HGP cells in L, M and H PD status, respectively. Telomeres were elongated with only a short delay of some PDs after immortalization, indicating that telomerase efficiently and rapidly elongated the telomeres.

Parallel to the telomere elongation, the average *hTERT* mRNA expression level was markedly increased from low PDs onwards and remained on a high level from L to H PDs (Figure 3). This was observed for both the absolute and relative *hTERT* mRNA levels at L, M, and H PD status. The degree of TL elongation did not correlate to the amount of mRNA expression in HGP and control cells (Figure 4). By contrast, we observed a weak (albeit not significant) inverse correlation between *hTERT* mRNA and TL in primary control (r_s_ = −0.448, *p* = 0.145) and HGP cells (r_s_ = −0.629, *p* = 0.016), which may point to the suppression of *hTERT* mRNA levels in cells with long telomeres, as previously proposed [30].

To better estimate the amount of *hTERT* mRNA needed to extend TL, we analyzed and compared the *hTERT* mRNA expression and the TL in each cell line. After immortalization, a decreased expression of *hTERT* mRNA in HGP188 was observed (680-fold change) at H PD, while TL continuously increased from L to H PD. By contrast, HGP003 failed to elongate telomeres despite a high expression of *hTERT* mRNA (2109-fold change) at H PD, suggesting that (i) the amount of *hTERT* expression *per se* is not the only factor for efficient telomere elongation and/or (ii) that other factors may also affect the telomere extension and/or (iii) under optimal conditions the amount of mRNA is far above the required amounts for full activity. We found several indications of the latter possibility. Among the control cells, the *hTERT*-infected N17 cell line had the lowest mRNA expression (395-, 106-, and 275- fold change at L, M, and H PD status, respectively) with a substantial degree of TL elongation (2.204, 2.282, and 1.782 T/S Ratio) compared to the primary cells before immortalization, suggesting that even a relatively low expression of *hTERT* mRNA (106-fold change) is sufficient to extend telomeres effectively. In none of the cell lines was the efficiency of TL elongation dependent on *hTERT* expression. (Appendix A)

### 3.3. Positive Influence of hTERT and Telomere Elongation on Cellular Stress Levels

The senescence-associated β-galactosidase activity (SA-β-gal) is considered to be a reliable biomarker of cellular senescence. The commonly used chromogenic substrate 5-Bromo-4-chloro-3-indolyl-β-D-galactopyranoside (X-gal) is often used to stain the cellular SA-β-gal activity. This methodology has, however, disadvantages, such as observer bias and lack of standardization. Therefore, we used a highly sensitive quantitative fluorimetric test—MUG fluorescence assay—to measure complete endogenous SA-β-gal activities in cell extracts.

Generally, we observed a continuous increase in senescence levels from L to H PD in primary HGP and control cells (Figure 5 and Appendix A). Immediately after infection (L PD after immortalization), four *hTERT* infected control cell lines (N14, N16, N17, and N20) and three *hTERT* infected HGP cell lines (HGP164, HGP178, and HGP188) had higher SA-β-gal activities, suggesting that the retroviral transduction may temporarily increase cell stress (Appendix A). At high PDs, however, the average SA-β-gal values of all virus-infected HGP and control cells were significantly reduced compared to primary cells in high PD (*p* < 0.016) (Figure 5), suggesting an overall “therapeutic” effect.

### 3.4. Similarities and Differences between Control and HGP Cells

We did not observe significantly increased telomere erosion in primary HGP cells compared to control cells (Appendix A). In both primary control and HGP cells, the TL continuously decreased from L to H PDs (Figure 2). By contrast, TLs immediately increased after *hTERT*-infection (Figure 2). The TL of three HGP cell lines (HGP127, HGP164, and HGP188) and four control cell lines (N14, N16, N18, and N20) gradually increased from L to H PD or remained constant at elevated levels after immortalization (Appendix A). Both control and HGP *hTERT*-expressing cells were continuously cultured for at least 50 PDs without a reduction in proliferative capacity.

We noticed, however, some subtle differences between control and HGP cell lines. Two of the HGP fibroblasts lines (HGP164, HGP188) showed substantially delayed telomere elongation: one cell line (HGP003) displayed very moderate telomere elongation (Appendix A). By contrast, all five control cell lines displayed a substantial degree of telomere elongation after *hTERT* infection, and only one cell line (N14) showed a slightly delayed response (Appendix A). Moreover, HGPS cells were more severely affected by the retroviral infection procedure. The control cells with empty vectors senesced slightly earlier, whereas the HGPS cells with empty vector died a few PDs after infection (Appendix A). Overall, the HGPS cells seem to be more vulnerable, which makes immortalization itself more difficult, keeping in mind the limitation that the higher immortalization PD in HGP cells may have influenced the phenotype (see also limitations in Section 4). Interestingly, we observed a weak negative correlation between TL and *hTERT* mRNA expression, which was significant for primary HGP fibroblasts (*p* = 0.016; Figure 4).

## 4. Discussion

The induction of telomerase activity by introducing the *hTERT* gene is a highly efficient method for the immortalization and rejuvenation of HGP cells. The immortalization of cells with *hTERT* has many potential advantages over other immortalization methods. Fibroblasts immortalized with SV40 large T antigen may accumulate chromosomal rearrangements, whereas fibroblasts immortalized with *hTERT* retain the control karyotype [31], which enables studies with less artificially changed cells.

The here described findings further support this hypothesis. By successful immortalization of all cell lines, we demonstrated the usefulness of the method to extend the cell life span. The immortalization with *hTERT* allowed the production of sufficient amounts of cell material for further experiments, which has also been shown in other studies, especially for methods with substantial cell material requirements [32]. In particular, the phenotype of diseased cells can be studied independently of increased aging under artificial *in vitro* conditions, which is extremely helpful for rare diseases [25,32,33]. Moreover, the comparison between primary and *hTERT* immortalized cells may also help to characterize age-specific changes caused by telomere shortening and to identify novel anti-aging strategies. Finally, the induction of *hTERT* using suitable gene transfer methods is a therapeutic option in HGPS, which must first be tested well in vitro. In this context, there is currently little data on the kinetics and the process of immortalization in relation to phenotype improvement, and some peculiarities could be pointed out in this study.

### 4.1. Initial Telomere Stabilization Is Important for Long Lasting Replication Induction and Phenotype Improvement

Ectopic expression of *hTERT* effectively elongated telomeres and protected HGP cells from premature senescence. Remarkably, the average TL substantially increased within the first 20 PDs after the infection. The TL only slightly increased or even decreased at higher PDs (Appendix A), while cell division rates accelerated from PD 20 onwards. These results indicate that the main elongation effect induced by *hTERT* occurs during the early period after immortalization, and that this initial telomere stabilization and elongation leads to a long-lasting replication induction, the prevention of senescence and phenotype improvement.

Apparently, all further cell growth was independent of further telomere elongation. Individual cell lines displayed a rapid proliferation rate with only moderate further telomere elongation or even TL reduction, suggesting that telomeres were long enough for replication and that the TL *per se* does not play a major role in cell growth. It has previously been shown that the shortest telomere, not the average TL, is critical for cell viability [34]. The data presented here also suggest that this stabilization of the telomere cap (and less the total length of the telomeres) is responsible for the positive effects on cell growth and phenotype. Moreover, additional functions of *hTERT* may modulate cell viability. For example, *hTERT* binds to the end of telomeres and may thus stabilize the telomere structure and promote cell proliferation independently from telomere elongation [35]. In accordance with this concept, abnormal telomere structure and the occurrence of individual short telomeres rather than the overall (average) telomere shortening was shown to be responsible for the induction of replicative senescence [17,36].

### 4.2. Phenotype Improvement Is Visible in Control and HGP Fibroblasts but Displays Substantial Delay

Replicative senescence is associated with increased cellular stress. We were therefore interested in the extent to which and when cellular stress is reduced by telomere lengthening. The cellular amount of SA-β-gal as a marker of cellular stress and senescence induction could be normalized by *hTERT* in both control and HGP cells. However, in most cell lines (4/5 HGP, 3/5 controls), the effect was not detectable at low and medium PDs after immortalization. Thus, the reduction in cellular stress occurred with a substantial delay and was not visible within the first replication cycles after immortalization with already significant growth promotion. Several reasons may be accounted for this delay in phenotype improvement. First, the repair mechanisms may require a relatively long period of time and several cell cycle runs. Second, it is possible that not the telomere elongation *per se*, but secondary effects are responsible for the cell stabilization. Third, the immortalization procedure may cause temporal cell stress that may have superimposed possible stabilizing effects. The latter possibility is supported by the observation that control cells and particularly the HGPS cells with empty vectors senesced much earlier than the primary cells without empty vectors. This can also be the cause for the relatively long period of growth arrest after infection (up to 100 days in culture in one HGP cell line), pointing to a critical phase, particularly in the diseased cells.

Avoiding cell stress is especially important for HGP cells. The altered lamin structure may affect the rate of telomere attrition and may cause other disturbances that lead to accelerated replicative senescence in HGP. In our experiments the average TL was not decreased in HGP cells, which is in contrast to decreased TL in another study [15]. However, the reported TL difference was rather moderate. In fact, normal or even longer telomeres have been described in approximately one-third of all cells despite shorter average TLs [15,22], suggesting an abnormal TL distribution in HGP. In addition, it has to be taken into account that our controls were obtained by older donor patients (since it is difficult to obtain control cells from healthy children) and therefore might have slightly shorter telomeres, so that a disease-dependent difference may have remained undetected. It is likely that HGP cells have a higher proportion of critically shortened or instable telomeres [15,22]. In this context, it is of interest that *hTERT* prefers the extension of the shortest telomeres, which may contribute to the longer growth arrest phase in HGP immortalization despite similar *hTERT* expression levels in controls and HGP cells. Overall, our results show that the expression of *hTERT* can improve the proliferation of all HGP cells, and—with some delay—can also reduce the levels of a key senescence marker. Data support the assumption that *hTERT* is a possible therapeutic approach for HGP, using genetic engineering technologies [37]. This is in accordance with recently published results showing that transient expression of *hTERT* significantly stabilizes the phenotype of HGP cells [23].

### 4.3. Phenotype Improvement Was Not Dependent on the Amount of hTERT mRNA and the Degree of Telomere Elongation

The TL was not positively related to *hTERT* expression, and the senescence reduction (measured by use of the senescence marker SA-β-gal) occurred independently from the TL and *hTERT* expression. Several reasons may be accounted for this phenomenon. First, a relatively low amount of *hTERT* mRNA might be sufficient for telomere elongation and other protective effects, which has important implications for therapeutic strategies. It is possibly less important to induce large amounts of *hTERT* mRNA than to use gentle methods of gene transfer. The latter aspect might be of relevance in view of the apparent stress effects by the immortalization procedure. Second, positive effects on the phenotype might be caused by secondary effects or by effects of *hTERT* independent from telomere elongation. Telomerase has various potentially protective effects on cell functions besides telomere elongation. For example, disturbed DNA-damage signaling is mainly located at the telomere [38,39] and telomerase may modulate progerin-induced DNA-damage signaling [24]. Mitochondrial dysfunction decreases ATP levels and increases ROS in HGP cells [40,41] associated with swollen and broken mitochondria [42]. *hTERT* has a mitochondrial targeting sequence at the N-terminus [43] and *hTERT* is able to maintain mtDNA integrity by increasing the mitochondrial membrane potential and reducing mitochondrial superoxide and ROS levels [44].

In HGP primary cells, we found rather a weak inverse correlation between the *hTERT* mRNA expression and the TL. It is tempting to speculate, but must be examined in detail in other experiments, that this effect may be caused by telomere position effects such as telomere position effect over long distances (TPE-OLD), e.g., the suppression of gene transcription by interference with long telomeres [27], and that this process might be disturbed in HGP [45].

### 4.4. Limitations

This study has several limitations. First, we investigated only a limited number of fibroblast cell lines. The incidence of HGP is low and there are ethical concerns regarding skin biopsies in healthy children. To minimize confounders, we used skin biopsies from HGP children with the most frequent mutation only. Second, the control probands were older (since it was impossible to obtain control cells from healthy children under local conditions). This and the higher immortalization PD in HGP cells may have influenced phenotype differences before and after immortalization. In particular, differences in the primary cells (in TL and SA-β-GAL activity) could be underestimated in HGP cells.

These pitfalls should be avoided in follow-up studies. However, these limitations do not undermine the main results of the study, especially with regard to the immortalization kinetics data. Findings are internally consistent and can help to optimize immortalization conditions for therapeutic studies.

## 5. Conclusions

In conclusion, ectopic expression of *hTERT* mRNA is a powerful tool for immortalizing HGP fibroblasts. This could help to supply sufficient cell material for further research and to transform senescent HGP cells to a younger, healthier state. Telomere elongation mainly occurred in the early stage of immortalization. Telomerase is able to elongate the telomeres and to stabilize the cellular phenotype. Phenotype improvement under the commonly used in vitro conditions is independent from the amount of cellular *hTERT* mRNA, the degree of telomere elongation, and occurs with substantial delay. Additional experiments are needed to further explore the factors responsible for the protective effects of *hTERT*.

## Figures and Tables

**Figure 1 cells-11-02784-f001:**
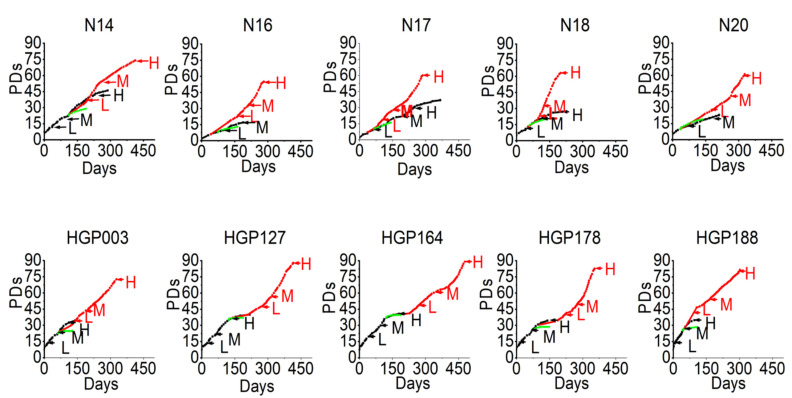
The growth curves of control (N) and HGP fibroblasts. Primary fibroblasts in black, immortalized fibroblasts in red, negative control cells (infected with empty vector) in green. L, low PD; M, medium PD. H, high PD.

**Figure 2 cells-11-02784-f002:**
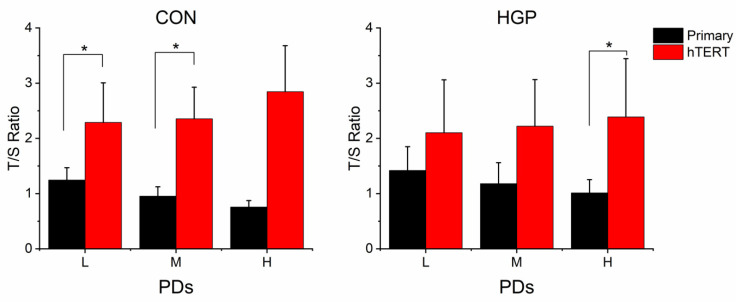
The average telomere length of control (CON) and HGP cells before and after immortalization. TL as T/S ratio (ratio of the content of telomere (T) to a single-copy gene (S)). HGP (*n* = 5) and control (*n* = 5) cells at L, M, and H PD. The data are presented as the mean ± SD and analyzed using Mann–Whitney-U-Test. * *p* ≤ 0.05, Bonferroni correction *p* < 0.016 (*hTERT* cells vs. primary cells at the corresponding PD class). L, low PD; M, medium PD. H, high PD.

**Figure 3 cells-11-02784-f003:**
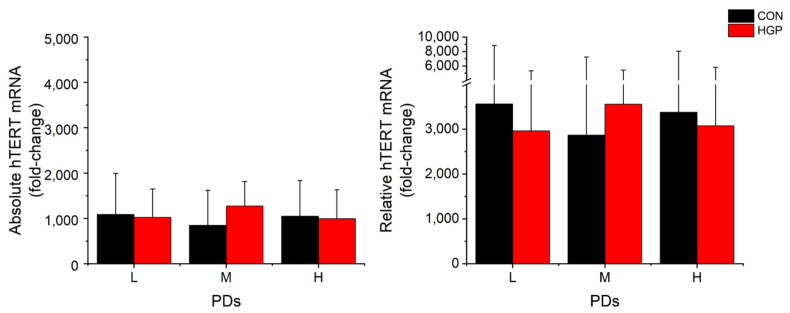
Average absolute and relative *hTERT* mRNA (fold-change) expression of immortalized fibroblasts. The absolute and relative mRNA expression of each cell line is shown for control and HGP fibroblasts at L, M, and H PD, as described in Materials and Methods. The data are presented as the mean ± SD and analyzed using Mann–Whitney-U-Test. There was no significant difference (HGP vs. CON at the corresponding PD class). L, low PD; M, medium PD. H, high PD.

**Figure 4 cells-11-02784-f004:**
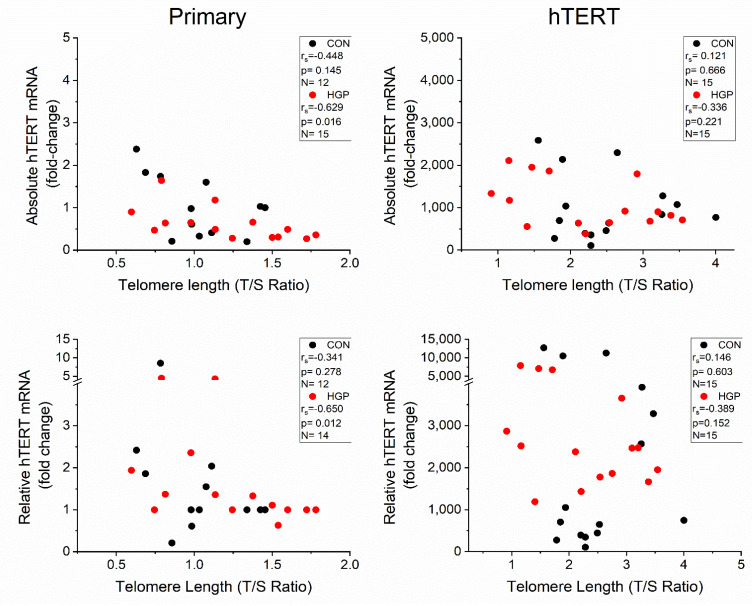
Scatter plot for the absolute and relative mRNA (fold-change) vs. telomere length (T/S Ratio) of primary and *hTERT* control (black) and HGP (red) cells. The absolute and relative mRNA expression was calculated as described in Materials and Methods. Spearmans rho and p values are presented. T/S ratio represents the ratio of the content of telomere (T) to a single-copy gene (S) and represents the overall relative telomere length.

**Figure 5 cells-11-02784-f005:**
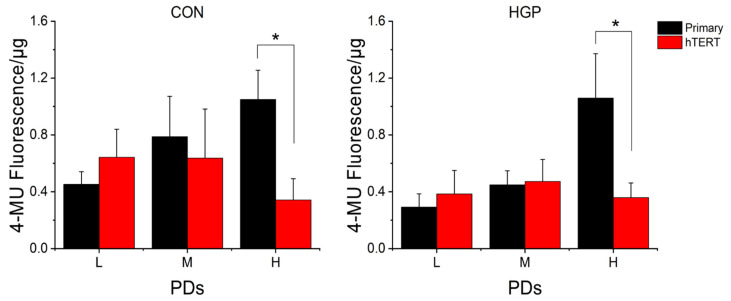
Average SA-β-gal activity (4-MU fluorescence) of healthy control (CON) and HGP primary cells (black) and immortalized cells (red) in different PDs. The intracellular stress levels of immortalized HGP (*n* = 5) and control (*n* = 5) cells (from L to H PDs) were compared with their primary cells, respectively. The data are presented as the mean ± SD and analyzed using Mann–Whitney-U-Test. * *p* ≤ 0.05, Bonferroni correction *p* < 0.016 (*hTERT* vs. primary cells at the corresponding PD class). L, low PD; M, medium PD. H, high PD.

**Table 1 cells-11-02784-t001:** HGP and control cell lines used in this study.

Cell Line	Cell Line for Short	Age, Sex	Molecular Defect	Retroviral Infection PD
HGADFN003	HGP 003	2 years 0 month, male	Exon 11 mutation C->T, clinically affected	23.6
HGADFN127	HGP 127	5 years 0 month, female	Exon 11 mutation C->T, clinically affected	37
HGADFN164	HGP 164	4 years 8 months, female	Exon 11 mutation C->T, clinically affected	39.2
HGADFN178	HGP 178	6 years 11 months, female	Exon 11 mutation C->T, clinically affected	30.6
HGADFN188	HGP 188	2 years 3 months, female	Exon 11 mutation C->T, clinically affected	27
Control Fibroblast	N14	50–60 years, female	No	24
Control Fibroblast	N16	50–60 years, female	No	6
Control Fibroblast	N17	50–60 years, female	No	7
Control Fibroblast	N18	50–60 years, Female	No	13
Control Fibroblast	N20	50–60 years, female	No	10

## Data Availability

Not applicable.

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
