# Peer review of "Establishment and Characterization of hTERT Immortalized Hutchinson–Gilford Progeria Fibroblast Cell Lines"

_cells, 2022, doi:10.3390/cells11182784_

Round 1

Reviewer 1 Report (Previous Reviewer 1)

The authors established immortalized fibroblast cell lines from patients with Hutchinson–Gilford progeria and evaluated the effects of hTERT immortalization on proliferative life span and senescence. They observed a phenotypic improvement which was found not to be dependent on telomere elongation. The topic is of high interest.

This manuscript has been resubmitted after previous evaluation and I found that the authors implemented the suggestions I made on the first version. The introduction has been revised in order to include relevant information on the studied disorder as well as on telomeres. A limitation paragraph has been added in the discussion. The authors increased the sample size of the control group compared to the previous version. In addition, the methods section has been integrated with relevant information to allow a more comprehensive evaluation of this study. 

I found the statistical section as well as the description of results to still lack some information. For instance, have the data shown in the barplots in Figure 1 and 2 been analyzed with a statistical test or are they simply descriptive?  At page 5, lines 206-207, the following sentence should be revised as correlation analysis and linear regression are two different statistical methods: “Correlation analysis of TL and mRNA expression was carried out by linear regression using Graphpad Prism 9.” In addition, the authors should add whether they checked assumptions of the used statistical methods, and how. 

In Figure 3, the figure should be revised in order to show the comparisons that have actually been made (reporting the bars that were compared next to each other and not in different graphs). 

Author Response

Reviewer 1

The authors established immortalized fibroblast cell lines from patients with Hutchinson–Gilford progeria and evaluated the effects of hTERT immortalization on proliferative life span and senescence. They observed a phenotypic improvement which was found not to be dependent on telomere elongation. The topic is of high interest.

This manuscript has been resubmitted after previous evaluation and I found that the authors implemented the suggestions I made on the first version. The introduction has been revised in order to include relevant information on the studied disorder as well as on telomeres. A limitation paragraph has been added in the discussion. The authors increased the sample size of the control group compared to the previous version. In addition, the methods section has been integrated with relevant information to allow a more comprehensive evaluation of this study. 

I found the statistical section as well as the description of results to still lack some information. For instance, have the data shown in the barplots in Figure 1 and 2 been analyzed with a statistical test or are they simply descriptive?  At page 5, lines 206-207, the following sentence should be revised as correlation analysis and linear regression are two different statistical methods: “Correlation analysis of TL and mRNA expression was carried out by linear regression using Graphpad Prism 9.”

In addition, the authors should add whether they checked assumptions of the used statistical methods, and how. 

In Figure 3, the figure should be revised in order to show the comparisons that have actually been made (reporting the bars that were compared next to each other and not in different graphs). 

Response:

We thank the reviewer for the positive overall evaluation.

To address the accurate comments on the statistical parts, we moved the descriptive raw data of the individual cell lines into the supplement. This applies for parts of figure 1 and 2, and for former figure 6.

To provide a better fitting test for correlation analysis, we replaced the linear regression by Spearman’s rho analysis, since not all assumptions for linear regression were applicable. We used nonparametric tests only, because the sample size for each group was not big enough to be based on a normal distribution .

To provide a better comparison, former figure 3 (now figure 2) was revised.

Reviewer 2 Report (Previous Reviewer 2)

The revised version of the manuscript was improved by the inclusion of missing controls. Furthermore, important aspects, such as study limitations, were added in the Discussion section. However, data presentation and statistics remain incorrect (comparing data presented on different graphs by Mann-Whitney test, using a statistical test for two samples several times instead of ANOVA or at least correcting the p-value for multiple testing). Moreover, the same data are presented several times:

-        - Figure 3 shows the analysis of telomere length presented in Figure 1 and Figure 2 (average vs individual data for each cell line)

-        - Table 2 shows the exact same data as Figure 3 but in a different format

-       - Figure 4 shows the analysis of hTERT mRNA presented in Figure 1 and Figure 2 (average vs individual data for each cell line)

-       - Figure 7 shows the analysis of B-gal activity presented in Figure 6 (average vs individual data for each cell line)

While it is a good practice to show the raw data for each cell line individually, it should be done in a supplementary figure. Showing and analyzing the same data on different figures/tables can give a false impression that more information is given when it is not.

Furthermore, the conclusions are limited by the use of cells from 50–60-year-old individuals as controls. While I understand the difficulty in obtaining samples from healthy age-matched children, these controls could be obtained form young adults (20 years old). In my opinion, when the appropriate tools or materials are unavailable, then the experiments should not be performed since they cannot give any conclusive results. Overall, the manuscript was improved compared to the initial version; however, it still presents serious flaws. In particular, data presentation and analysis should be revised by a statistician.

Author Response

Reviewer 2

The revised version of the manuscript was improved by the inclusion of missing controls. Furthermore, important aspects, such as study limitations, were added in the Discussion section. However, data presentation and statistics remain incorrect (comparing data presented on different graphs by Mann-Whitney test, using a statistical test for two samples several times instead of ANOVA or at least correcting the p-value for multiple testing).

Moreover, the same data are presented several times:

      Figure 3 shows the analysis of telomere length presented in Figure 1 and Figure 2 (average vs individual data for each cell line)

      Table 2 shows the exact same data as Figure 3 but in a different format

      Figure 4 shows the analysis of hTERT mRNA presented in Figure 1 and Figure 2 (average vs individual data for each cell line)

      Figure 7 shows the analysis of B-gal activity presented in Figure 6 (average vs individual data for each cell line)

While it is a good practice to show the raw data for each cell line individually, it should be done in a supplementary figure. Showing and analyzing the same data on different figures/tables can give a false impression that more information is given when it is not.

Furthermore, the conclusions are limited by the use of cells from 50–60-year-old individuals as controls. While I understand the difficulty in obtaining samples from healthy age-matched children, these controls could be obtained form young adults (20 years old). In my opinion, when the appropriate tools or materials are unavailable, then the experiments should not be performed since they cannot give any conclusive results. Overall, the manuscript was improved compared to the initial version; however, it still presents serious flaws. In particular, data presentation and analysis should be revised by a statistician.

Response:

We thank the reviewer for the thorough evaluation. We agree that changes are necessary to provide an improved version of the manuscript, even if the experimental approach (which we would not repeat in this way) remains limited in its validity, as explained in detail in the discussion section and below.

We changed the presentation of data. Except for the growth curves we moved all raw data into the supplement (T/S ratios, mRNA expressions of former figure 1 and 2, and former figure 6). Table 2 was deleted since it did not give any new information. We decided however to provide the growth curves for all cell lines since it improves the understanding of what we did and visualizes the differences between low, middle and high PDs before and after immortalization.

Former figure 3 was revised to improve the understanding of which conditions were tested against each other. We also tried to address your comment on p-values for multiple testing. Because of the low sample size, we did not perform ANOVA, since not all requirements were applicable. We decided to use a Bonferroni correction, which led to a required significance level of p<0.016. This was applied to new Figures 2 and 5. Figure 3 did not show any significance level below p<0,05.

We also revised former figure 5, now figure 4, since it was not the best way to present a correlation. We used Spearman’s rho to address this issue in the revised version, as suggested by a statistician expert.

Regarding your last comment, we agree that the choice of control cells was not ideal and further experiments are obligatory. We discuss this issue in detail in the Discussion section and added one sentence. Nevertheless, we think that the main findings are correct, even if some differences in primary cells (HGP versus controls) may have been underestimated due to the age of the controls. We feel that these limitations do not undermine the main results of the study, especially with regard to the immortalization kinetics data.

Reviewer 3 Report (Previous Reviewer 3)

The authors have adequately addressed my concerns. A nice study that should be of great interest to the HGPS and senescence research community.

Author Response

Reviewer 3

The authors have adequately addressed my concerns. A nice study that should be of great interest to the HGPS and senescence research community.

Response:

We thank the reviewer for the positive overall evaluation.

Round 2

Reviewer 2 Report (Previous Reviewer 2)

The revised version of the manuscript was significantly improved, especially in terms of statistical analysis. However, some minor issues need to be addressed before publication:

-        Data shown in Figure 3 (erroneously labelled as Figure 4, please correct) should be presented (and analyzed) in the same way as data in new figure 2 and 5

-        Please revise numbers throughout the manuscript (all decimal points should be shown with a period not with a comma, e.g. lines 256, 285, 292).

Author Response

Dear Reviewer,

Please find enclosed the revised version of the manuscript. Figure 3 was revised according to the reviewer’s suggestions.

Figure 3 was renamed, commas have been replaced where necessary.

We added an “Ethics and Informed Consent Statement” and one sentence in the “Acknowledgments” section.

We are looking forward hearing from you

Very Sincerely 

Michael Walter

This manuscript is a resubmission of an earlier submission. The following is a list of the peer review reports and author responses from that submission.

Round 1

Reviewer 1 Report

The authors investigated the effect of ectopic expression of the catalytic subunit of telomerase using hTERT-retrovirus infected Hutchinson–Gilford progeria fibroblasts and control fibroblast cells. The study showed increased proliferative life span and reduced number of senescent cells. Elongation of telomeres did now depend on the level of hTERT mRNA expression in HGP cells.

The topic is of interest and the article is well written and easy to read. The methods appears to be sound and results support the conclusions. I only have minor comments to improve clarity of some parts:

  • In the introduction, line 30, it would be useful to report the percentage of cases in which the mutation in the LMNA gene is present in patients with Hutchinson-Gilford progeria syndrome.
  • The statistical analysis paragraph should be integrated as the Mann-Whitney U test wasn't the only test used (for instance correlation analysis was used and it should be reported, together with the specific test used, e.g. Pearson or Spearman's correlation). It should also be reported for which variables each test was used.
  • In the same paragraph, the software and version used for the analyses should be added.
  • In the discussion, a paragraph listing limitations of the study should be added.

Reviewer 2 Report

In the current manuscript, Lin and colleagues generated five Hutchinson-Gilford progeria cell lines immortalized by ectopic expression of the catalytic subunit of telomerase (hTERT). The authors observed that TERT-immortalized progeria fibroblasts showed increased proliferative lifespan and reduced senescence. The manuscript lacks sufficient conceptual novelty as TERT-expressing progeria fibroblast were previously described and characterized, for example:

  • Kudlow BA, Stanfel MN, Burtner CR, Johnston ED, Kennedy BK. Suppression of proliferative defects associated with processing-defective lamin A mutants by hTERT or inactivation of p53. Mol Biol Cell. 2008 Dec;19(12):5238-48. doi: 10.1091/mbc.e08-05-0492. Epub 2008 Oct 8. PMID: 18843043; PMCID: PMC2592682.
  • Benson EK, Lee SW, Aaronson SA. Role of progerin-induced telomere dysfunction in HGPS premature cellular senescence. J Cell Sci. 2010 Aug 1;123(Pt 15):2605-12. doi: 10.1242/jcs.067306. Epub 2010 Jul 6. PMID: 20605919; PMCID: PMC2908049.
  • Li Y, Zhou G, Bruno IG, Cooke JP. Telomerase mRNA Reverses Senescence in Progeria Cells. J Am Coll Cardiol. 2017 Aug 8;70(6):804-805. doi: 10.1016/j.jacc.2017.06.017. PMID: 28774385.
  • Li Y, Zhou G, Bruno IG, Zhang N, Sho S, Tedone E, Lai TP, Cooke JP, Shay JW. Transient introduction of human telomerase mRNA improves hallmarks of progeria cells. Aging Cell. 2019 Aug;18(4):e12979. doi: 10.1111/acel.12979. Epub 2019 May 31. PMID: 31152494; PMCID: PMC6612639.

Furthermore, the study is poor, and it presents serious flaws in its design. For instant:

  • Control fibroblasts originate from 50–60-year-old individuals instead of age-matched children.
  • There are only 3 control fibroblast lines; given the variability between different lines, it is necessary to include more samples in the analysis to be able to draw any conclusions from the study.
  • Retroviral infection was performed at 23-39 passage doublings for progeria cells, and at 7-13 for control cells. Some of the differences observed between control and progeria cells may arise from this.
  • No empty virus control was included in the study; given that a retroviral vector was used for the delivery of hTERT, insertional mutagenesis might be contributing to some effects observed in the study.

Moreover, data is inappropriately presented and analyzed, e.g., the authors for statistical analysis compare data presented on different graphs (see Figures 4/5/6).

Reviewer 3 Report

The authors provide the most detailed analysis of the potential role of telomere length maintenance to date. Previous studies have shown the exogenous expression of hTERT can ameliorate the reduced proliferative capacity of HGPS fibroblasts in culture, but had not analyzed hTERT expression levels or telomere length comprehensively. I commend the authors for examining multiple cell lines and for their perseverance in taking on such a time consuming study. This work will be an aid to those studying HGPS which has been somewhat limited by the inability to grow patient cells for long term studies.

My only suggestion for improvement would be the addition of a short section in the discussion on the potential role for an hTERT effect on mitochondria. Mitochondrial disfunction has been implicated in HGPS, primarily from the work of Kan Cao's lab, but also others. Also, hTERT has been shown to be involved in mitochondrial functions. I realize that this study focuses on telomere elongation as the causative effect, but a brief mention of a potential mitochondrial role as an adjuvant to the improvement of the cells is in order.